# Acute kidney disease stage predicts outcome of patients on extracorporeal membrane oxygenation support

**Cheng-Kai Hsu**[1], **I-Wen Wu**[1,2], **Yih-Ting Chen**[1], **Tsung-Yu Tsai**[3], **Feng-Chun Tsai**[2,4], **Ji-Tseng Fang**[2,3], **Yung-Chang Chen**[1,2,5]*

**1** Department of Nephrology, Chang Gung Memorial Hospital, Keelung, Taiwan, **2** College of Medicine, Chang Gung University, Taoyuan, Taiwan, **3** Department of Nephrology, Chang Gung Memorial Hospital, Linkou, Taiwan, **4** Division of Cardiovascular Surgery, Chang Gung Memorial Hospital, Linkou, Taiwan, **5** Community Medicine Research Center, Keelung Chang Gung Memorial Hospital, Keelung, Taiwan

* cyc2356@gmail.com

**Data Availability Statement:** All relevant data are within the paper and its Supporting Information files.

## Abstract

### Background

The mortality rate of patients on extracorporeal membrane oxygenation (ECMO), especially those patients that develop acute kidney injury (AKI) is high. Acute kidney disease (AKD) is a term used to describe the continuum from AKI to chronic kidney disease. However, the role of AKD in predicting the prognosis of patients on ECMO support is unclear.

### Methods

A total of 168 patients who received ECMO support and survived for more than 7 days at a single hospital from 2003 to 2008 were enrolled for this study and followed up for 10 years or till mortality. Kaplan-Meier analysis and Cox proportional hazards model were used to determine the prognostic factors associated with survival.

### Results

The median survival times of patients with stage 0, stage 1, stage 2 and stage 3 AKD were $\geq$ 10 years, 43.9 months, 1 month, and half a month, respectively. There were statistically significant differences in cumulative survival rate between patients with stage 3 AKD and those with stage 0, 1, and 2 AKD (Cox-Mantel log rank test, $p < 0.001$, $p < 0.001$, $p = 0.023$), and between patients with stage 0 AKD and those with stage 1 and 2 AKD (Cox-Mantel log rank test, $p = 0.012$, $p < 0.001$). Cox regression analysis revealed that AKD stage (hazard ratio [HR]: 2.576, 95% confidential interval [CI]: 1.268–5.234, $p = 0.009$ for stage 1; HR: 2.349; 95% CI: 1.101–5.512, $p = 0.029$ for stage 2; HR: 5.252; 95% CI: 2.715–10.163, $p < 0.001$ for stage 3) was significant independent predictor of survival.

### Conclusion

AKD stage is an independent predictor of survival in patients on ECMO support.

**Funding:** The authors received no specific funding for this work.

**Competing interests:** The authors have declared that no competing interests exist.

## Introduction

Extracorporeal membrane oxygenation (ECMO) is often used for critically ill patients with respiratory and cardiac failure, and it may reduce the risk of progressive organ dysfunction. It can be used as a bridge-to-recovery, bridge-to-transplant, or bridge-to-decision. However, the in-hospital mortality rate of patients on ECMO supports is high at 21%-58% depending on the indications for the intervention [1–6]. Furthermore, the overall mortality rate is as high as 26% in patients who successfully wean from ECMO [7]. Urine output is an independent predictor of in-hospital mortality [4, 7]. The outcomes of patients on ECMO support worsen when acute kidney injury (AKI) occurs in the course of their disease.

The global burden of AKI is around 13.3 million cases per year with hospitalizations for AKI rising over time. A published meta-analysis showed that in the United States alone, AKI is associated with 1 hospitalization every 7.5 minutes [8]. A single-center study reported that the in-hospital mortality rate and 6-month survival rate of patients treated with ECMO who had concomitant AKI were worse than those of patients who did not develop AKI [6]. A recent meta-analysis study also reported that the incidence rate of AKI in patients receiving ECMO is as high as 62.8%, and that in-hospital mortality is 3.7 times higher in patients on ECMO who develop AKI that require renal replacement therapy [9]. Another previously published meta-analysis showed that the incidences of AKI and severe AKI in pediatric patients who require ECMO are high [10]. However, the severity of AKI is also an important prognostic factor for the outcome of patients on ECMO support. The cumulative survival rate of patients treated with ECMO who have stage 3 AKI according to the Kidney Disease: Improving Global Outcomes (KDIGO) classification is worse than that of patients treated with ECMO who have KDIGO stage 1 and stage 2 AKI [6].

The transition and continuum of AKI to chronic kidney disease (CKD) remain a growing area of investigation [11]. AKI and CKD share an inter-correlated bidirectional pathway and many risks and prognostic factors [12, 13]. The incidence rates of CKD and end-stage renal disease (ESRD) after an episode of AKI were reported to be 7.8 events/100 patient-years and 4.9 events/100 patient-year, respectively [14]. According to several meta-analysis studies, AKI is an independent risk factor of CKD (adjusted hazard ratio [HR]: 8.8) and ESRD (HR: 3.1) [15, 16]. However, clinical outcome remains unclear in patients with acute kidney disease (AKD), a clinical condition defined as the presence of KDIGO stage 1 AKI or greater occurring or persisting in the 7–90 days after an initial AKI event (Fig 1). Stage 1 AKD is defined as an increase of serum creatinine level to 1.5–1.9 times, stage 2 AKD is defined as an increase of serum creatinine level to 2.0–2.9 times, and stage 3 AKD is defined as an increase of serum creatinine level to $\geq 3.0$ times the baselines levels that occurred or persisted in 7–90 days after renal injury [11]. Furthermore, the exact roles of AKD classification in prognosis in patients on ECMO support remain largely unclear.

In this study, we aimed to investigate the relationship between cumulative survival rate and AKD classification in patients on ECMO support.

## Materials and methods

### Patient characteristics and settings

This study was conducted in accordance with the Declaration of Helsinki and approved by the Institutional Review Board (IRB) at Chang Gung Memorial Hospital (IRB-201901629B0). The need for consent was waived by the ethics committee. This retrospective cohort study recruited patients admitted to the intensive care unit (ICU) of Chang Gung Memorial Hospital who

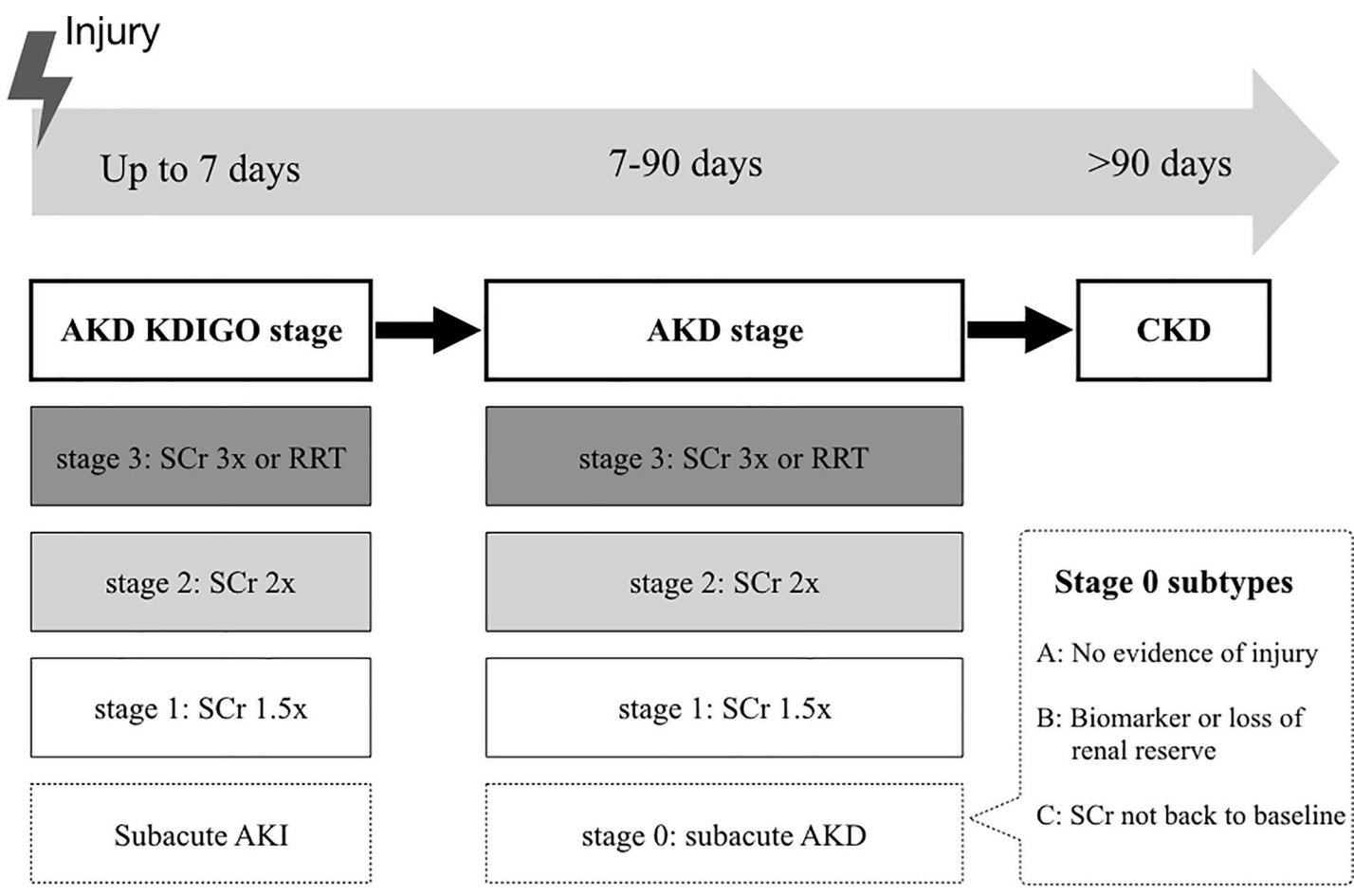

**Fig 1. Interplay between AKI, AKD and CKD according to the consensus report of the Acute Disease Quality Initiative (ADQI) 16 workgroup.** AKD denotes acute or subacute loss of renal function for a duration of 7–90 days after exposure to an AKI initiating event, and it can progress to CKD. Stage 0 AKD represents partial recovery from AKI. Abbreviations: KDIGO, Kidney Disease: Improving Global Outcomes; RRT, renal replacement therapy; SCr, serum creatinine.

received ECMO support from August 1, 2003 to December 31, 2008. Patients treated with ECMO who died within 7 days of hospitalization (92 patients), patients with ESRD requiring dialysis therapy at baseline (14 patients), and patients younger than 18 years (38 patients) were excluded from this study. Finally, a total of 168 patients were followed for 10 years from the initiation of ECMO (Fig 2). For patients who required repeated ECMO support during the study period, only data from the first ECMO treatment were collected for analysis.

The patients were divided into AKD and non-AKD group according to the criteria of AKD stage 1 to stage 3 since the initiation of ECMO support. Data including demographic and biochemical data, primary diagnosis for ICU admission, and AKD classification were collected for further analysis. The worst physiological parameters of day 1 of ECMO support were recorded for analysis. The study endpoint was all-cause mortality in the follow-up period up to 2018.

### Definitions

The incidence and severity of AKD were evaluated based on the definition of the consensus report of the Acute Disease Quality Initiative (ADQI) 16 workgroup. The index date was defined as day 1 of ECMO support. AKI and AKD were diagnosed by specialists in

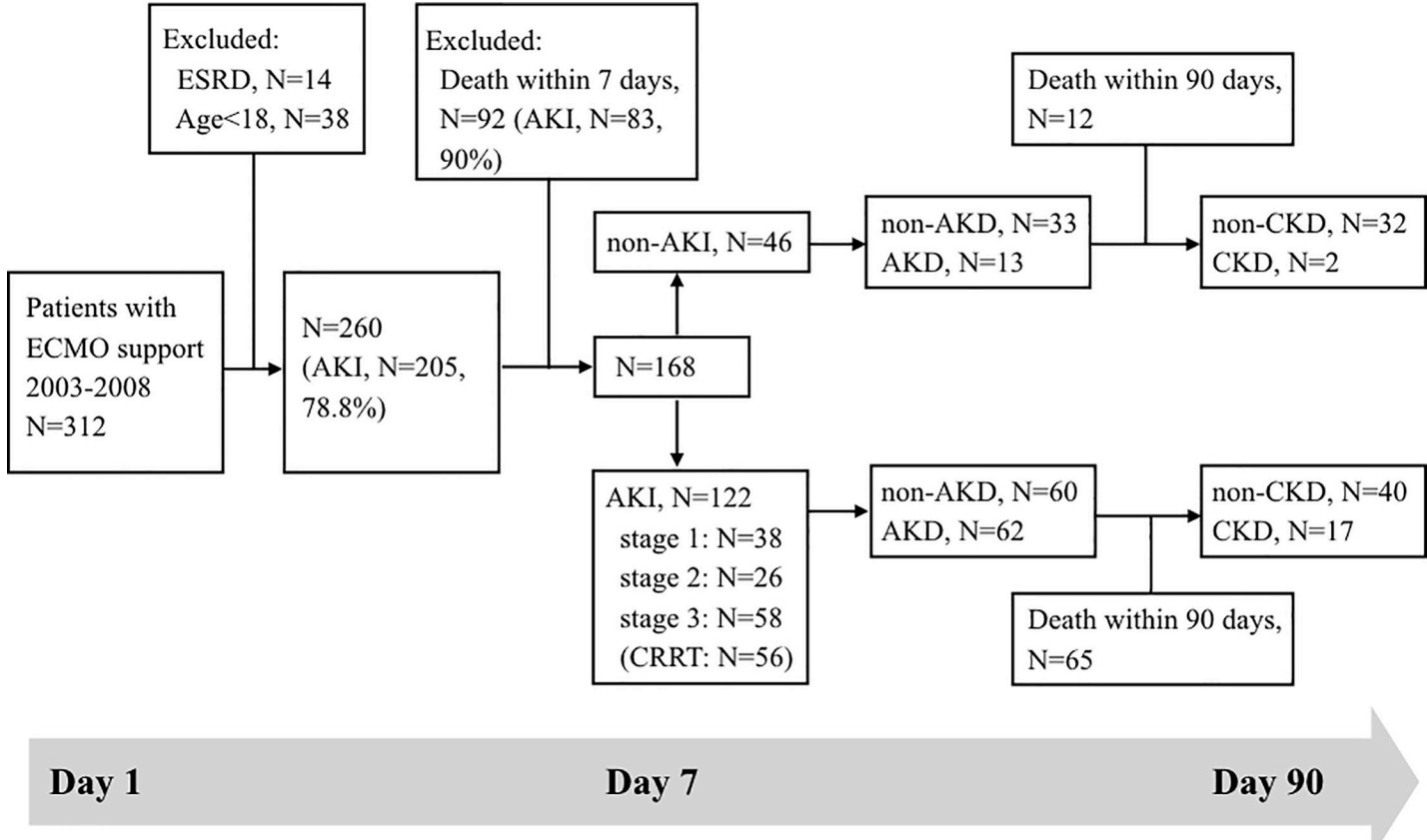

**Fig 2. Study design flowchart.** Abbreviations: AKD, acute kidney disease; AKI, acute kidney injury; CKD, chronic kidney disease; CRRT, continuous renal replacement therapy; ECMO, extracorporeal membrane oxygenation; ESRD, end-stage renal disease.

nephrology. We determined the estimated glomerular filtration rate (eGFR) using the CKD epidemiology collaboration equation. AKI was defined according to the KDIGO criterion of renal failure that developed within 7 days of initiation of ECMO, and AKD was defined as a clinical condition characterized by increase of serum creatinine that occurred or persisted between 7 and 90 days after initiation of ECMO support. The highest serum creatinine level in the first 7 days was used for staging AKI and the highest serum creatinine level in 7–90 days was used for staging AKD. Stage 1 AKD was defined as an increase of serum creatinine level to 1.5–1.9 times the baseline level, stage 2 AKD was defined as an increase of serum creatinine level to 2.0–2.9 times the baseline level, and stage 3 AKID was defined as an increase of serum creatinine level $\geq 3.0$ times the baseline level. Patients with an increase of serum creatinine level to less than 1.5 times the baseline level between 7 and 90 days after initiation of ECMO support were categorized as the non-AKD groups (stage 0 AKD). Baseline serum creatinine level was defined as the lowest serum creatinine level in the 30 days preceding ECMO support or the lowest serum creatinine level in the 30 days following ECMO support if serum creatinine levels were not measured in the 30 days before ECMO support.

## Clinical management

The ECMO device (Medtronic, Inc., Anaheim, CA) included a centrifugal pump and a silicone oxygenator (Medtronic, Minneapolis, MN, USA) with an integrated heater. A heparin-bound

Carmeda bioactive surface was used in the ECMO circuit. Peripheral cutaneous cannulation with a closed sternum was performed, and the cut-down procedure was used for some obese patients. We used a 17–19 French percutaneous arterial (infusion) cannula and a 19–21 French percutaneous venous (drainage) cannula (DLP; Medtronic Inc., Minneapolis, MN). For veno-arterial ECMO, percutaneous access was preferred via the common femoral artery (infusion) and the common femoral vein (drainage).

### Statistical analysis

Descriptive statistics were expressed as mean ± standard deviation or median (interquartile range). Discrete variables were presented as frequencies and percentages. The Kolmogorov-Simirnov method was used to test the normality of the numerical variables. Differences in continuous variables between the AKD and non-AKD groups were analyzed using the Student's t-test or Mann Whitney U test. Differences in categorical variables between the 2 groups were compared using the chi-square test or Fisher's exact test. Analysis of survival was derived from the Kaplan-Meier analysis. The Cox proportional hazards model was used to identify the prognostic factors associated with mortality. Multivariate logistic regression analyses were conducted to determine the combination of risk factors for AKD. Statistical analysis was performed using the Statistical Package for the Social Sciences (SPSS) version 21.0 (SPSS, Inc., USA). All statistical tests were 2-tailed, and a $p$-value $<0.05$ was considered statistically significant.

### Results

From August 1, 2003 to December 31, 2008, a total of 168 patients who had received ECMO support and survived for more than 7 days were enrolled in this study, and they consisted of 120 patients without CKD at baseline and 48 patients with CKD at baseline (stage 3: N = 36, stage 4: N = 7, stage 5: N = 5). There were 93 patients without AKD and 75 patients with AKD. Of the 75 patients with AKD, 30, 13, and 32 patients had stage 1, stage 2, and stage 3 AKD, respectively. The mean age of all the patients was 53.1 ± 16.2 years and 112 (66.7%) of them were men. Table 1 summarizes the demographic data and clinical characteristics of the patients. Patients with AKD were older, had longer durations of ECMO support, and were less likely to wean from ECMO support than patients without AKD. The AKD group had lower Glasgow Coma Scale (GCS) score, lower mean arterial pressure (MAP), lower urine output, lower blood platelet count, higher serum K concentration, higher Acute Physiology and Chronic Health Evaluation (APACHE) II score, and higher Sequential Organ Failure Assessment (SOFA) score than the non-AKD group. There was no statistically significant difference in baseline eGFR between the 2 groups, and there were no differences in other biochemical parameters, such as serum albumin and serum lactate levels, on day-1 of ECMO support.

Table 2 shows the primary diagnoses of patients who require ECMO support and the types of ECMO (veno-arterial or veno-venous). The 2 most common indications for ECMO support in our study were post-cardiotomy cardiogenic shock (42.9%) and acute respiratory distress syndrome (28.6%). There were no statistically significant differences in primary diagnosis and type of ECMO between the 2 groups.

The median survival times of patients with stage 0, stage 1, stage 2 and stage 3 AKD were $\geq$ 10 years, 43.9 months, 1 month, and 16 days, respectively. There were statistically significant differences in cumulative survival rate between patients with stage 3 AKD and those with stages 0, 1, and 2 AKD (Cox-Mantel log rank test, $p<0.001$, $p<0.001$, $p = 0.023$, Fig 3) and between patients with stage 0 AKD and those with stage 1 and stage 2 AKD (Cox-Mantel log rank test, $p = 0.012$, $p<0.001$, Fig 3). Cox regression analysis revealed that age, initial SOFA

**Table 1. Demographic data and clinical characteristics of study population.**

|  | All patients (n = 168) | Non-AKD (n = 93) | AKD (n = 75) | *p*-value |
|---|---|---|---|---|
| Age (years) | 53.1 ± 16.2 | 50.4 ± 15.8 | 56.4 ± 16.1 | 0.016 |
| Men, number (%) | 112 (66.7) | 58 (62.4) | 54 (72.0) | 0.249 |
| Duration of ECMO support, median (IQR), (days) | 7 (4–12) | 6 (3–12) | 9 (5–15) | 0.024* |
| Weaning from ECMO support, number (%) | 130 (77.4) | 82 (88.2) | 48 (64.0) | <0.001 |
| GCS score on day 1 of ECMO, median (IQR) | 14 (8–15) | 14 (9–15) | 11 (6–14) | 0.024* |
| MAP on day 1 of ECMO (mmHg) | 56.2 ± 18.0 | 59.1 ± 19.7 | 52.6 ± 14.8 | 0.018 |
| UO on day 1 of ECMO, median (IQR), (mL) | 1643 (626–2607) | 1872 (950–2955) | 1250 (360–2332) | 0.008* |
| Baseline eGFR (mL/min/1.73m$^2$) | 76.6 ± 30.9 | 76.1 ± 30.7 | 77.1 ±31.3 | 0.827 |
| WBC count on day 1 of ECMO (cells/mL x 1000) | 14.4 ± 8.8 | 14.7 ± 8.1 | 14.1 ± 9.6 | 0.675 |
| Hemoglobin concentration on day 1 of ECMO (g/dL) | 9.2 ± 2.1 | 9.4 ± 2.4 | 9.0 ± 1.7 | 0.276 |
| Platelet count (1000 cells/mL x 1000) | 96 (54–146.5) | 97 (70–163) | 82 (41.5–141.5) | 0.021* |
| Na$^+$ concentration on day 1 of ECMO (mEq/L) | 146.6 ± 7.8 | 147.0 ± 7.5 | 146.1 ± 8.2 | 0.455 |
| K$^+$ concentration on day 1 of ECMO (mEq/L) | 3.9 ± 1.1 | 3.8 ± 1.1 | 4.2 ± 1.1 | 0.020 |
| Albumin concentration on day 1 of ECMO (g/dL) | 2.6 ± 0.7 | 2.7 ± 0.7 | 2.6 ± 0.6 | 0.920 |
| Bilirubin concentration on day 1 of ECMO (mg/dL) | 2.6 ± 4.8 | 1.9 ± 2.5 | 3.5 ± 6.5 | 0.051 |
| Lactate concentration on day 1 of ECMO (mg/dL) | 82.2 ± 58.8 | 76.6 ± 55.8 | 89.1 ± 62.0 | 0.226 |
| AaDO$_2$ on day 1 of ECMO (mmHg) | 357.8 ± 209.8 | 349.1 ± 209.0 | 368.5 ± 211.7 | 0.554 |
| APACHE II score on day 1 of ECMO, median (IQR) | 22 (17–29) | 19 (15–26) | 25 (20–30) | 0.001* |
| SOFA score on day 1 of ECMO | 12.0 ± 4.0 | 11.4 ± 3.6 | 12.7 ± 4.3 | 0.031 |

Abbreviations: AKD, acute kidney disease; ECMO, extracorporeal membrane oxygenation; IQR, interquartile range; GCS, Glasgow Coma Scale; MAP, mean arterial pressure; UO, urine output; eGFR, estimated glomerular filtration rate; WBC, white blood cell; AaDO$_2$, alveolar-arterial oxygen difference; APACHE, Acute Physiology and Chronic Health Evaluation; SOFA, Sequential Organ Failure Assessment

* *p*-value using Mann-Whitney U test.

score, urine output on day 1 of of ECMO support, and AKD stage (HR: 2.576, 95% CI: 1.268–5.234, *p* = 0.009 for stage 1; HR: 2.349, 95% CI: 1.101–5.512, *p* = 0.029 for stage 2; HR: 5.252, 95% CI: 2.715–10.163, *p*<0.001 for stage 3) were significant independent predictors of survival

**Table 2. Primary diagnosis of patients admitted to the ICU who require ECMO support and types of ECMO.**

|  | All patients n (%) | Non-AKD n (%) | AKD n (%) | *p*-value |
|---|---|---|---|---|
| Post-cardiotomy cardiogenic shock | 72 (42.9) | 38 (40.9) | 34 (45.3) | 0.639 |
| Acute respiratory distress syndrome | 48 (28.6) | 26 (28.0) | 22 (29.3) | 0.844 |
| Acute myocardial infarction | 13 (7.7) | 7 (7.5) | 6 (8.0) | 1.000* |
| Myocarditis | 12 (7.1) | 10 (10.8) | 2 (2.7) | 0.068* |
| Decompensated heart failure | 8 (4.8) | 5 (5.4) | 3 (4.0) | 0.733* |
| Shock status simultaneous with hypoxic respiratory failure | 4 (2.4) | 1 (1.1) | 3 (4.0) | 0.865* |
| Sudden collapse status post CPCR | 4 (2.4) | 1 (1.1) | 3 (4.0) | 0.325* |
| Severe traumatic injury | 4 (2.4) | 4 (4.3) | 0 (0) | 0.129* |
| Post-lung or heart transplantation | 3 (1.8) | 1 (1.1) | 2 (2.7) | 0.587* |
| Types of ECMO |  |  |  | 0.654 |
| Veno-arterial | 116 (69) | 66 (71) | 50 (66.7) |  |
| Veno-venous | 52 (31) | 27 (29) | 25 (33.3) |  |

Abbreviations: AKD, acute kidney disease; ECMO, extracorporeal membrane oxygenation; CPCR, cardiopulmonary cerebral resuscitation; ICU, intensive care unit.

* *p*-value determined using Fisher's exact test.

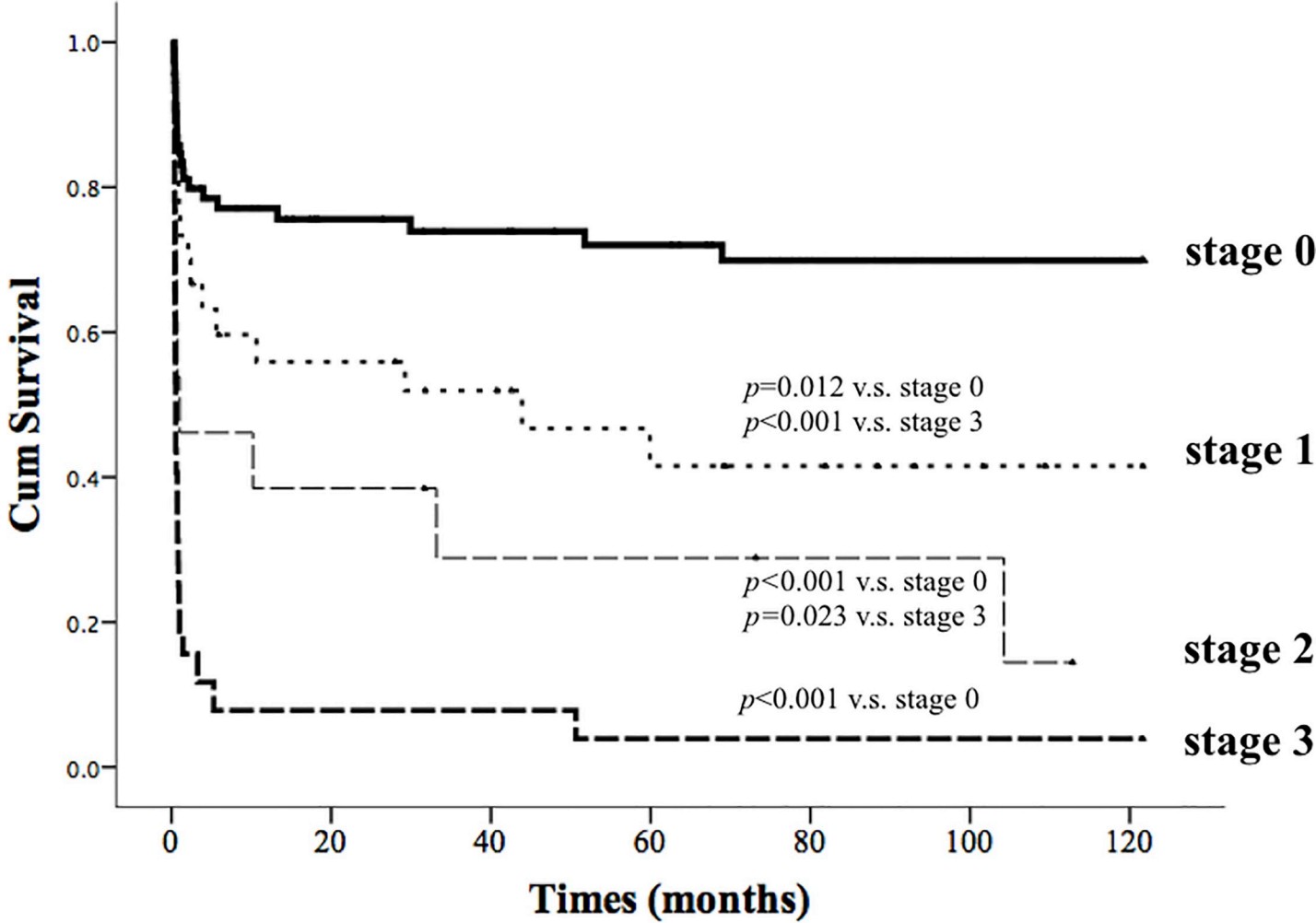

**Fig 3. Cumulative survival rate of 168 critically ill patients on extracorporeal membrane oxygenation support according to the stages of acute kidney disease.**

after adjusting for sex, age, duration of ECMO support, APACHE II score, SOFA score, GCS score, MAP, urine output, baseline eGFR, and AKD stage (Table 3).

The odds ratios of the variables for the prediction of AKD determined using logistic regression are shown in Table 4. Urine output and serum potassium concentration on day 1 of ECMO support were found to be significant independent risk factors of AKD after adjusting for age, sex, duration of ECMO support, APACHE II score, SOFA score, GCS score, MAP, urine output, serum potassium level, and baseline eGFR.

## Discussion

In this retrospective cohort study, we showed that stage of AKD is an independent risk factor of mortality in patients receiving ECMO support. To the best of our knowledge, this study is the first to show the association between AKD and poor prognosis in patients receiving ECMO treatment. Our study highlight the importance and clinical utility of AKD classification in predicting the prognosis of patients treated with ECMO support.

The roles of age, urine output, KDIGO stage, GCS score, and duration of ECMO support in increasing the mortality of patients receiving ECMO therapy are well established [1, 3, 4, 6,

**Table 3. Hazard ratios for the mortality of patients on ECMO.**

| Variables | Crude | | Adjusted* | |
|---|---|---|---|---|
| | HR (95% CI) | *p*-value | HR (95% CI) | *p*-value |
| Age (years) | 1.031 (1.015–1.047) | <0.001 | 1.037 (1.016–1.057) | <0.001 |
| Duration of ECMO support (days) | 1.020 (1.008–1.032) | 0.001 | 1.004 (0.989–1.02) | 0.57 |
| APACHE II score on day 1 of ECMO | 1.062 (1.036–1.089) | <0.001 | 1.031 (0.982–1.082) | 0.223 |
| SOFA score on day 1 of ECMO | 1.223 (1.150–1.302) | <0.001 | 1.241 (1.129–1.364) | <0.001 |
| GCS score on day 1 of ECMO | 0.919 (0.877–0.963) | <0.001 | 1.017 (0.929–1.114) | 0.717 |
| MAP on day 1 of ECMO (mmHg) | 0.988 (0.976–1.002) | 0.083 | 1.001 (0.984–1.017) | 0.947 |
| UO on day 1 of ECMO (100mL) | 0.949 (0.929–0.969) | <0.001 | 0.968 (0.945–0.991) | 0.007 |
| K on ECMO first day (mEq/L) | 0.952 (0.629–1.443) | 0.818 | 1.071 (0.867–1.323) | 0.527 |
| Baseline eGFR (ml/min/1.73m$^2$) | 0.997 (0.99–1.004) | 0.44 | 1.007 (0.998–1.016) | 0.131 |
| AKD stage | | | | |
| Stage 0 | 1 | | 1 | |
| Stage 1 | 2.237 (1.187–4.214) | 0.013 | 2.576 (1.268–5.234) | 0.009 |
| Stage 2 | 3.859 (1.843–8.079) | <0.001 | 2.349 (1.101–5.512) | 0.029 |
| Stage 3 | 9.069 (5.182–15.872) | <0.001 | 5.252 (2.715–10.163) | <0.001 |

Abbreviation: ECMO, extracorporeal membrane oxygenation; APACHE, Acute Physiology and Chronic Health Evaluation; SOFA, Sequential Organ Failure Assessment; GCS, Glasgow Coma Scale; MAP, mean arterial pressure; UO, urine output; eGFR, estimated glomerular filtration rate; AKD, acute kidney disease

* Adjusted for gender, age, duration of ECMO support, APACHE II score, SOFA score, GCS score, MAP, urine output, baseline eGFR, and AKD stage.

17]. A study reported that daily urine output, MAP, and SOFA score are independent predictors of in-hospital mortality in critically ill patients successfully weaned from ECMO [7]. Similarly, our findings show that old age, high SOFA score, and low urine output on day 1 of ECMO support are associated with an increased risk of mortality. However, our study also found that long duration of ECMO support, GCS score, and MAP on day 1 of ECMO support were not predictors of mortality in patients on ECMO support after adjusting for age, duration of ECMO support, APACHE II score, SOFA score, GCS score, MAP, urine output, serum

**Table 4. Risk factors in a multivariate logistic regression model for AKD in patients on ECMO.**

| Variables | Crude | | Adjusted* | |
|---|---|---|---|---|
| | OR (95% CI) | *p*-value | OR (95% CI) | *p*-value |
| Age (years) | 1.024 (1.004–1.045) | 0.018 | 1.027 (0.997–1.058) | 0.158 |
| Male gender | 1.552 (0.805–2.990) | 0.189 | 1.107 (0.487–2.518) | 0.808 |
| Duration of ECMO support (days) | 1.040 (1.002–1.079) | 0.037 | 1.039(0.995–1.084) | 0.082 |
| APACHE II score on day 1 of ECMO | 1.055 (1.015–1.097) | 0.007 | 1.001 (0.928–1.081) | 0.972 |
| SOFA score on day 1 of ECMO | 1.091 (1.007–1.181) | 0.033 | 0.962 (0.837–1.106) | 0.59 |
| GCS score on day 1 of ECMO | 0.915 (0.852–0.982) | 0.014 | 0.87 (0.761–1.002) | 0.077 |
| MAP on day 1 of ECMO (mmHg) | 0.977 (0.959–0.997) | 0.022 | 0.979 (0.956–1.003) | 0.083 |
| UO on day 1 of ECMO (100mL) | 0.967 (0.945–0.990) | 0.005 | 0.969 (0.941–0.997) | 0.033 |
| K$^+$ concentration on day 1 of ECMO (mEq/L) | 1.390 (1.047–1.845) | 0.023 | 1.482 (1.015–2.163) | 0.042 |
| Baseline eGFR | 1.001 (0.991–1.011) | 0.826 | 1.010 (0.996–1.023) | 0.153 |
| AKI | 2.623 (1.26–5.462) | 0.01 | 2.257 (0.781–6.518) | 0.133 |

Abbreviation: ECMO, extracorporeal membrane oxygenation; APACHE, Acute Physiology and Chronic Health Evaluation; SOFA, Sequential Organ Failure Assessment; GCS, Glasgow Coma Scale; MAP, mean arterial pressure; UO, urine output; eGFR, estimated glomerular filtration rate

* Adjusted for age, sex, duration of ECMO support, APACHE II score, SOFA score, GCS score, MAP, urine output, serum potassium concentration, and baseline eGFR.

potassium concentration, and AKD stage (Table 3). The differences in primary diagnosis for ICU admission and the exclusion of patients who survived for less than 7 days may have contributed to the results of our study.

AKI is an independent risk factor of CKD [15,16]. In our study, univariate analysis showed that AKI is a significant risk factor (HR: 2.623, $p = 0.01$) of AKD among patients receiving ECMO support. However, there is no statistical significance in multivariate analysis after adjusting for age, sex, duration of ECMO support, APACHE II score, SOFA score, GCS score, MAP, urine output, serum potassium level, and baseline eGFR (HR: 2.257, $p = 0.133$). The reason may be attributed to the relative small size of our study.

The potential mechanisms underlying the progression of AKI to CKD include maladaptive repair (vascular dropout, tubular dropout, fibrosis, and unopposed transforming growth factor-β), pre-existing CKD with relatively low renal reserves, proteinuria leading to loss of nephron mass and hyperfiltration, and mitochondrial dysregulation [12, 15]. We observed a gradual increase in risk of mortality with increase in AKD stage in patients on ECMO support. Patients with stage 3 AKD had the highest mortality risk, and this emphasized the importance of kidney protection in improving outcomes of patients who require ECMO support. The intervention of ameliorating AKD severity may be similar to management with patients with CKD. Renin-angiotensin system blockers, sodium-glucose cotransporter 2 inhibitors, and anti-inflammatory agents may play a role.

Further studies are necessary to investigate the effects of increasing AKD severity on outcomes of patients on ECMO and the management of amelioration of AKD severity.

There are several limitations in our study. First, the study was conducted at a single tertiary care medical center, and the sample size was relatively small. Second, in-hospital mortality rate was as high as 38% in our study, especially in the early years. Consequently, even though the total follow-up duration was 10 years, the mean and median follow-up durations were only 1060 and 146.5 days, respectively. Third, AKI biomarkers were not measured, and this may have led to underestimation of patients with stage 0 as we categorized patients with AKD stage 0 as the non-AKD group. In addition, the prognosis of our patients was mainly based on baseline data and day 1 of ECMO. It is unclear if we use of periodic repeated measurements of these parameters added value to the prediction. Fourth, other variables such as quantity of intravenous fluid, volume balance, transfusion, intravenous contrast medium use, duration of renal replacement therapy, baseline medication, and baseline comorbidities were not measured in our study. Further large studies on patients receiving ECMO who develop AKD should consider taking these variables into account. Fifth, other insults that cause renal injury may have occurred in the 7–90 days. This is a potential limitation not only in our study, but also in other retrospective articles regarding AKI/AKD. Finally, although the primary diagnosis of patients for ICU admission did not differ between the 2 groups, the heterogeneity of the study population may limit the extrapolation of our findings to a single disease entity. Further studies on a large cohort of patients with a unique critical illness or from an ethnic group are needed to confirm our findings. Despite these drawbacks, the long follow-up period, coverage of important clinical risk factors, and comparison of multiple scoring systems for predicting outcomes in critical patients have strengthened the conjecture of this study.

In conclusion, AKD staging, defined by the ADQI 16 workgroup, is associated with poor survival rate in patients who require ECMO support due to various primary diseases, independent of sex, age, duration of ECMO support, APACHE II score, SOFA score, GCS score, MAP, urine output, baseline eGFR, and serum potassium level on day 1 of ECMO support. AKD staging may help in risk stratification of critically ill patients on ECMO support. Further studies with larger sample sizes and independent critical disease entities should be conducted to confirm the results of our study.

Our findings provide a basis for clinical application of AKD stage as a predictor of mortality in patients on ECMO who survive for more than 7 days. Due to the sample size in our study, it is necessary to conduct further studies in the future to confirm our findings.

## Supporting information

**S1 File. Raw data used for this manuscript.**
(XLSX)

## Acknowledgments

The authors thank the staff members at the ICU of Chang Gung Memorial Hospital for their assistance with patient management and data collection.

## Author Contributions

**Conceptualization:** I-Wen Wu, Feng-Chun Tsai, Yung-Chang Chen.

**Data curation:** Cheng-Kai Hsu, I-Wen Wu, Yih-Ting Chen, Tsung-Yu Tsai, Yung-Chang Chen.

**Formal analysis:** Cheng-Kai Hsu, I-Wen Wu, Yih-Ting Chen, Tsung-Yu Tsai, Ji-Tseng Fang, Yung-Chang Chen.

**Investigation:** Cheng-Kai Hsu, Yih-Ting Chen, Feng-Chun Tsai, Ji-Tseng Fang.

**Methodology:** Cheng-Kai Hsu, I-Wen Wu, Yih-Ting Chen, Tsung-Yu Tsai, Feng-Chun Tsai, Ji-Tseng Fang, Yung-Chang Chen.

**Project administration:** Cheng-Kai Hsu, I-Wen Wu, Yung-Chang Chen.

**Resources:** Feng-Chun Tsai.

**Supervision:** Cheng-Kai Hsu, Ji-Tseng Fang.

**Validation:** Cheng-Kai Hsu, I-Wen Wu, Tsung-Yu Tsai, Ji-Tseng Fang, Yung-Chang Chen.

**Visualization:** Cheng-Kai Hsu.

**Writing – original draft:** Cheng-Kai Hsu, I-Wen Wu, Yih-Ting Chen, Yung-Chang Chen.

**Writing – review & editing:** Cheng-Kai Hsu, I-Wen Wu, Tsung-Yu Tsai, Feng-Chun Tsai, Ji-Tseng Fang.

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
