## [Decision Letter · Decision Letter 0]

13 Jan 2020

PONE-D-19-31910

Acute Kidney Disease Stage Predict Outcome of Patients Receiving Extracorporeal Membrane Oxygenation Support- A 10-year Cohort Study

PLOS ONE

Dear Dr Chen,

Thank you for submitting your manuscript to PLOS ONE. After careful consideration, we feel that it has merit but does not fully meet PLOS ONE’s publication criteria as it currently stands. Therefore, we invite you to submit a revised version of the manuscript that addresses the points raised during the review process.

ACADEMIC EDITOR: 

The manuscript is of potential interest, especially because of the 10-year follow-up. However, it is not acceptable for publication in its current form. I believe the manuscript would benefit from providing more detailed both methodology and results description. For that reason I would ask you to provide the following information:

What was the type of ECMO (VV vs VA) used?How long were patients on ECMO support? How did this differ between the different subgroup of patients, and did the mortality outcomes differ when this was adjusted for?Could you please provide a clear sequential story od kidney failure i.e.the incidence of AKI in the first seven dayshow many patients developed severe AKI stage 3the duration of AKIhow many were treated with CRRT for how long.how many with AKI go on to qualify as AKDhow many across the 90 day mark to qualify as CKD/whayt was AKI recovery ratePerhaps in the form of a flowchart?

The mauscript would benefit from language editing.

There are also other minor issues, which are described in detail by the Reviewers.

We would appreciate receiving your revised manuscript by Feb 27 2020 11:59PM. To enhance the reproducibility of your results, we recommend that if applicable you deposit your laboratory protocols in protocols.io, where a protocol can be assigned its own identifier (DOI) such that it can be cited independently in the future. For instructions see: http://journals.plos.org/plosone/s/submission-guidelines#loc-laboratory-protocols

We look forward to receiving your revised manuscript.

Kind regards,

Justyna Gołębiewska

Academic Editor

PLOS ONE

2. Please provide additional details regarding participant consent. In the ethics statement in the Methods and online submission information, please ensure that you have specified (1) whether consent was suitably informed and (2) what type you obtained (for instance, written or verbal). If your study included minors under age 18, state whether you obtained consent from parents or guardians. If the need for consent was waived by the ethics committee, please include this information.

3. We noticed you have some minor occurrence(s) of overlapping text with the following previous publication(s), which needs to be addressed:

https://doi.org/10.1371/journal.pone.0202781

In your revision ensure you cite all your sources (including your own works), and quote or rephrase any duplicated text outside the Methods section. Further consideration is dependent on these concerns being addressed.

Reviewers' comments:

Reviewer's Responses to Questions

**Comments to the Author**

1. Is the manuscript technically sound, and do the data support the conclusions?

Reviewer #1: Partly

Reviewer #2: Yes

Reviewer #3: Yes

Reviewer #4: Partly

Reviewer #5: Yes

Reviewer #6: Yes

2. Has the statistical analysis been performed appropriately and rigorously? 

Reviewer #1: Yes

Reviewer #2: I Don't Know

Reviewer #3: Yes

Reviewer #4: I Don't Know

Reviewer #5: Yes

Reviewer #6: Yes

3. Have the authors made all data underlying the findings in their manuscript fully available?

Reviewer #1: Yes

Reviewer #2: Yes

Reviewer #3: Yes

Reviewer #4: Yes

Reviewer #5: No

Reviewer #6: Yes

4. Is the manuscript presented in an intelligible fashion and written in standard English?

Reviewer #1: Yes

Reviewer #2: No

Reviewer #3: No

Reviewer #4: Yes

Reviewer #5: Yes

Reviewer #6: Yes

5. Review Comments to the Author

Reviewer #1: It is an interesting and relevant article in the field of Nephrology and Critical Care. I consider it a useful contribution in its field, with the major strength of this study, which is long follow-up time (10 years). I have important comments for the authors to additionally improve their study/manuscript.

1. Of 168 patients receiving ECMO, 75 (44.6%) patients had acute kidney injury/disease, which is relatively lower incidence compared to overall AKI incidence of AKI in patients on ECMO. PMID: 31284451 doi: 10.3390/jcm8070981. Please discuss if this is because of different of patient population? VV or VA-ECMO.

2. In results page 16, Line 143, please additionally describe that of 75 patients with AKD, how many patients developed severe AKI stage 3. Majority of AKI in patients on ECMO is severe.

3. While it is already known that patients who develop AKI requiring RRT while on ECMO carry 3.7-fold higher hospital mortality (PMID: 31284451), your study has very long follow-up time, which is major strength of this study.

4. Any data on AKI recovery? At least among patients with severe AKI requiring renal replacement therapy/dialysis. How many had renal recovery and able to be off dialysis?

5. Indication for ECMO and type of ECMO (VV vs VA) should be taken into consideration.

Reviewer #2: This is an interesting single center study examining patients on ECMO having concomitant AKI and their associated mortality. This study is unique in that it specifically examines the outcomes of patients who survived to develop “acute kidney disease, AKD,” (as defined by Chawla et al.’s article as acute kidney injury between 7-30 days), and omits data on patients who developed AKI with mortality presenting within 7 days of ECMO and AKI. Based on the study methods, it is important to note that this study does not include patients who initiated ECMO and then developed AKI or AKD. Hence the initial renal injury preceded ECMO. I applaud and congratulate the authors on accomplishing a long-term study with novel results. There are however a few suggestions and recommendations that may improve the strength of this study.

Recommend additional editing for English grammar throughout the manuscript. For instance, grammar needs work through the manuscript to improvement readability. Other errors are small, but need correction such as page 4, line 52 where KDIGO is spelled incorrectly. In addition, page 4, line 51-54: the way it is currently written suggests that KDIGO states that AKIN severity impacts survival rates when I believe the authors intended to state that this reference 6 is what showed this finding.

Background: There is another recent meta-analysis published on impact of AKI on ECMO in J Clin Med in 2019 (PMID 31284451). The HR is similar to references already used, but this reference may be useful to include in order to provide the reader updated information.

Materials and Methods:

Can the authors confirm that this is a prospective study? Reviewing the methods, it sounds like a retrospective study. If prospective, how many patients were admitted to the ICU with ECMO support total (e.g. those without concomitant AKI)?

Was this VA or VV ECMO?

How long were patients on ECMO support? How did this differ between the different subgroup of patients, and did the mortality outcomes differ when this was adjusted for?

Page 7, line 103: this sentence is confusing. You previously noted that patients were recruited only if they were “treated with ECMO support having concomitant AKI.” In this sentence, you state that patients were divided into AKD and non-AKD groups if they met criteria of AKD stages 1-3 since initiation of ECMO support. I assume then that patients in the non-AKD consisted of patients who had initial AKI but no longer had AKI after 7 days (transitioned to AKD Stage 0A-C). If so, it may be worthwhile noting to the reader in the discussion section that the “non-AKD” subgroup is still at higher risk for worse outcomes than a patient who never developed AKI with subsequent transition to AKD Stage 0A-C.

Page 8, line 112: Was the standard definition of AKI Stage 1 utilized for inclusion into this study? This is important to specify so we understand what population we are looking at. The usual definition, as the authors undoubtedly know, includes not only the serum creatinine 1.5-1.9 times, but also an absolute increase in serum creatinine by ≥ 0.3 mg/dl or reduction in UOP to < 0.5 ml/kg/hr. Presumably, the non-AKD patients would include not only patients who developed AKIN Stage 1-3 who recovered, but also a subset of patients who had AKIN stage 1, but did not meet the serum creatinine baseline rise of 1.5-1.9x to reach AKD stage 1. If the authors only recruited patients on “ECMO with concomitant AKI” based on a Cr 1.5-1.9x rise, this excludes a significant number of patients. Please clarify.

Page 8, line 114: “The highest serum creatinine value was used for staging”. Over what time interval was the peak creatinine used for staging AKI? The authors note that only patients with concomitant ECMO and AKI were included, not patients who were initiated on ECMO and then developed AKI. So was the highest serum creatinine value the one that occurred within 24 hours of ECMO initiation? 6 hours? Etc.

Page 10, line 155: When/how was baseline creatinine determined?

Page 11: The authors did an excellent job including many clinical variables of interest including clinical socres associated with severity of baseline disease. Other variables that may impact outcomes include quantity of iv fluids, volume balance, transfusions, IV contrast, baseline comorbidities associated with AKI/CKD (e.g. diabetes, hypertension) would be useful to have included in the report. The prevalence of CKD and its severity would also be useful to understand the population being studied.

Page 13, line 173: What was the median survival time for AKD stage 0.

As urine output is related to AKI and used as a criteria for diagnosis and severity of AKI, it is not a surprise that if AKI/AKD was associated with mortality, then it would be significant as well. Hence, I wonder if urine output was not adjusted for, if it would impact the results. On a separate note, as noted above, perhaps if the total volume balance, it may help place the clinical impact of urine output (from volume balance versus its impact with AKI) in better context. Finally, as it is unknown if diuretics were utilized, the urine output and its clinical relevance is difficult to ascertain.

Page 15, line 193: As noted previously, urine output on first day of ECMO support as an independent risk factor for developing AKD seems of questionable relevance since urine output is used as a diagnostic criteria for AKI, and if you develop AKI, then automatically there is AKD stage 0 through 3. Also, I would recommend again to consider use of the estimated GFR (consider the CKI-EPI equation) as it incorporates a patient age, sex, and race to better estimate renal function. The eGFR can then not only be utilized for confounder adjustment, but it can be utilized to see if it predicts the occurrence and/or severity of AKD, or its impact on mortality.

Page 17: I agree that the findings of this study, including that patients requiring ECMO who developed AKD having worse survival, are novel. From a clinical standpoint, it would be valuable to know whether the worse outcomes is independently associated with AKD, or whether the presence of AKD is related to increased CKD, which thus translates to worse survival. It is well known that CKD is associated with increased comorbidities, quality of life, cardiovascular disease, and survival. Would a patient who developed AKD that resolved without developing CKD possess the same worse outcomes (e.g. is the time limited acute episode of ECMO and AKD enough to translate to long term outcomes?)

It would also be useful to know the survival times of patients who had ECMO without AKD for a comparison group, since this is a single center study in a specific population that may not be generalizable. As this study incorporated all ECMO patients, does AKD and ECMO stratified by primary diagnosis affect mortality outcomes? I do appreciate however that the authors already mention in the limitation paragraph of the discussion section that they were not able to stratify by primary diagnosis.

Page 16: How many AKD stage 3 patients requiring intermittent and/or continuous dialysis? The need for dialysis has been shown in prior studies to correlate with mortality.

Page 18, line 239: This may be due to how it is written, but it was not clear to me how pre-existing CKD cause progression of AKI to CKD?

Page 18: Line 235-236: Could you expound how the exclusion of patients who survived less than 7 days may contribute to the results? Did you mean to say that it would be a limitation to the study?

Page 18, line 246: “amelioration of AKD severity.” What does this mean. Regarding medical therapy? Dialysis? Novel future interventions like stem cell therapy or ischemic preconditioning?

Page 19, line 251: What doyou mean measurement of several AKI biomarkers was incomplete? This was the first time it was mentioned in the article. How many patients was data incomplete?

I think adding the following data would lend strength and interest to the study. Furthermore, the authors may already have this data. It would be interesting to compare the diagnosis and effects of AKI vs AKD vs resultant CKD with survival in ECMO patients, and see which one best predicted survival.

Reviewer #3: 1 “The study recruited patients admitted at the intensive care units (ICU) of Chang Gung Memorial Hospital and treated with ECMO support having concomitant AKI, classified according to KDIGO definition, from August 1, 2003, to December 31,2008.” Did you include only patients who developed AKI after ECMO? I don’t see the number of patients with no AKI that were excluded in the following statements. Did patients with AKD stage 0 need to have AKI before? Were you interested in AKI occurring before or after ECMO or both?

2. what was the time frame for assessment of AKD? Was it 7-90 days after initiation of ECMO? However, by the definition, AKD should be assessed within 7-90 days after AKI occurred?

3. study included patients from 2003 t0 2008. Although the follow-up time might up to 10 year, the wording” 10-year cohort study” is misleading. I don’t think you should use 10-year cohort study in this manuscript.

4. what is the definition of baseline creatinine

5. author should consider add “Incidence and Impact of Acute Kidney Injury in Patients Receiving Extracorporeal Membrane Oxygenation: A Meta-Analysis.” (PMID 31284451)

6. How did you obtain mortality outcome after hospital discharge

7. How did you select adjusting covariates

8. what is the time zero for survival analysis

9. manuscript need English editing

Reviewer #4: IN the manuscript, "Acute Kidney Disease Stage Predict Outcome of Patients Receiving Extracorporeal Membrane Oxygenation Support- A 10-year Cohort Study," the authors present prospective cohort study of 168 patients on ECMO. They present a novel evaluation of the concept of acute kidney disease and its impact on outcome. Overall the study is relatively large for the topic nd there is good data present. As it is currently presented though the study is hard to follow and the point is lost.

The authors should consider presenting the whole story of AKI to CKD. To me this would be a better story and make the point.

I would recommend describing the incidence of AKI in the first seven days. Then its impact on outcomes including mortality. Table 1

In this part it is critical to also evaluate the duration of AKI. This has become a critical point in the literature and is clearly associated with outcomes.

Table 2. Those that go on to qualify as AKD. How many with AKI go on to qualify as AKD. It is also critical to better define for the readers how you distinguish the two. Since you are using a novel idea this is a critical point. performs a flow diagram of those with AKI that go on to develop or qualify as AKD.

Table 3. Finally how many cross the 90 day mark to qualify as CKD. To me this is a potentially novel report in ECMO

A few critical details.

How many were treated with CRRT for how long. How were these classified in the staging. What is the practice for CRRT on ECMo at your institution?

What was the mortality for the whole cohort. This is a critical point for deciding the size of the models. To me this study may have too many variables in the regression analysis unless the mortality was quite high. Typically I am familiar with 1 variable for each ten events of interest (at least for standard multivariable analysis.)

Data should not be presented in table form and written out. Te results section could be shortened significantly by doing this.

Also were there tests of normality performed to decide median verse means for data presentation?

Reviewer #5: Kai Hsu et al., have submitted their findings of a 10-year cohort study on the ‘acute kidney disease stage predicts the outcome of the patients receiving the extracorporeal membrane oxygenation support.’

The study question is relevant, and the authors have done a good job in the design of the study and preparing and presenting the results.

Here are my comments,

1.Introduction: Furnish the epidemiology on the ECMO and AKI If possible.

-Needs to provide the definitions of the AKD. It’s not routinely used in the clinical practice and needs to furnish further evidence on its definition and staging with the citations. The authors cited ‘citation no.8. But requires to provide a citation from the professional nephrology society or guidelines if any available. Chawla et al. ( PMID: 28239173 ) wrote that ‘’As we have proposed new definitions, provided guidance for clinical practice and put forth a large agenda for future research, it will be incumbent on the AKI clinical research community to test our recommendations’’. I have not seen major literature on the topic in the last three years since the publication. I appreciate the authors shed some light as the definitions are not tested in extensive studies yet’.

Also, comment on the same in their ‘title.’

-Please include the following article findings in the introduction PMID: 31284451

2.Materials and Methods:

- Page 7. The authors need to furnish more data in the patient schema. Again, it’s ideal for defining the AKD criteria and mentioning how they were divided into stages 1-3 in the introduction.

-Under definitions: Please disclose how did the authors describe the ‘baseline’ renal function of the study subjects?

-What was the admission serum Creatinine in the subjects? were there any differences in the admission serum Cr/eGFR between AKD and Non-AKD patient groups?

- Were the study subjects on any medications which could potentially influence the serum creatinine?

- Clinical management: Keep it simple and mention if its veno arterial or venovenous ECMO or mixed cases of both at the beginning.

3.Discussion: The authors wrote, ‘For those patients under ECMO support, assessment of AKD staging should be mandatory, and it may help for the risk stratification and mortality prognosis in critical illness patients.

How does this will influence the clinicians in risk stratification and mortality prognosis in this very critically ill group of patients as in the majority of them, the ECMO is last and only chance to save their lives?

The authors should know that in a recent meta-analysis by Thongprayoon et al., In patients on the ECMO, the pooled estimated incidence of AKI and severe AKI requiring RRT were 62.8% (95%CI: 52.1%-72.4%) and 44.9% (95%CI: 40.8%-49.0%), respectively.

Reviewer #6: This is a well done study even though sample size is small which authors have pointed out but i believe this will serve as a foundation for similar large scale studies to be conducted. Below are my suggestions.

1) Insert following sentence in line 222. "A previous published meta-analysis shows that incidence of AKI and severe AKI in the pediatric population requiring ECMO is high". Please cite the following paper for this sentence.

Hansrivijit P, Lertjitbanjong P, Thongprayoon C, Cheungpasitporn W, Aeddula NR, Salim SA, Chewcharat A, Watthanasuntorn K, Srivali N, Mao MA, Ungprasert P. Acute Kidney Injury in Pediatric Patients on Extracorporeal Membrane Oxygenation: A Systematic Review and Meta-analysis. Medicines. 2019 Dec;6(4):109.

2) Please delete sentence “The AKD is an emerging medical entity in the field of critical care and the clinical significance of this kidney condition remains unraveled to date”. Don’t think this is necessary for flow of this paper.

3) KDIGO is spelled as KIDGO in some areas of paper. Please make necessary changes.

4) Please insert the following sentence anywhere in Introduction. “Global burden of AKI is around 13.3 million cases a year with hospitalizations for AKI rising over time. A published meta-analysis shows that in united states alone there is one hospitalization associated with AKI every 7.5 minutes". Please cite the paper as below.

Thongprayoon C, Kaewput W, Thamcharoen N, Bathini T, Watthanasuntorn K, Lertjitbanjong P, Sharma K, Salim SA, Ungprasert P, Wijarnpreecha K, Kröner PT. Incidence and Impact of Acute Kidney Injury after Liver Transplantation: A Meta-Analysis. Journal of clinical medicine. 2019 Mar;8(3):372.

5) Would prefer to use AKI instead of AKD throughout the paper

6) Wondering Why is that in logistic regression done by you, stages of AKI was not a significant risk factor? How do you explain that

7) Paper definetly needs some English language editing.

6. PLOS authors have the option to publish the peer review history of their article (what does this mean?). If published, this will include your full peer review and any attached files.

Reviewer #1: No

Reviewer #2: No

Reviewer #3: No

Reviewer #4: No

Reviewer #5: No

Reviewer #6: No

---

## [Author Response · Author response to Decision Letter 0]

20 Feb 2020

Answers to Editor:

1. The editor asked for information about types of ECMO. Among 168 patients, 116 patients received veno-arterial ECMO and 52 patients received veno-venous ECMO. We revised the Table 2 and added the description in the text. (Table 2, page 12, line 165-168)

2. The editor asked for the information about the duration of ECMO support and the differences between subgroups of patients. The median duration of ECMO support of all patients was 7 (IQR: 4-12) days, and AKD subgroup had longer duration of ECMO support (9 versus 6 days, p=0.024) than non-AKD subgroup. AKD stages were still significant independent predictors for survival after adjusting for duration of ECMO support. (Table 1, page 10, line 146; Table 3, page 13, line 181-183).

3. The editor suggested describing a clear sequential story of renal failure in the form of a flowchart. We modified Figure 2 according to editor’s valuable advice. The incidence of AKI was 78.8%. Among 168 patients, 58 patients developed AKI stage 3 and 56 patients received CRRT support with a median duration of 8 days. Among 122 patients with AKI, 62 patients developed AKD. Nineteen patients developed CKD 90 days later. Among 122 patients with AKI, 40 patients did not progress to CKD 90 days later. (Figure 2)

4. The editor asked for language editing. We are very sorry for our poor English. The manuscript had received English editing according to Editor’s suggestion.

Answers to Reviewer 1:

1. The reviewer proposed concerns on the focus of relative lower incidence of AKD/AKI. We modified Figure 2, and demonstrated that the incidence of AKI was 78.8% among 260 ECMO patients. Among patients who survived more than 7 days, the incidence of AKD was 44.6%. (Figure 2)

2. The reviewer asked for the information about the number of patients developed AKI and AKD stage 3. Among 75 patients with AKD, 30, 13, and 32 patients were AKD stage 1, stage 2, and stage 3 respectively. Among 122 patient with AKI, 58 patients developed severe AKI stage 3. We added the information to the Figure 2 and the text. (page 11, line 145-146)

3. The reviewer provided important recent study about the ECMO patients who develop AKI requiring RRT carry 3.7-fold higher hospital mortality (PMID:31284451). We added this reference and modified the introduction section. (page 4, line 50-52) 

4. The reviewer asked for information about AKI recovery. We modified Figure 2 to try to describe a clear sequential story of renal failure in the form of a flowchart. Among 122 AKI patients, 60 patients did not develop AKD and 40 patients did not develop CKD. Among 56 severe AKI patients who received CRRT and survived more than 90 days, 3 patients were able to be off dialysis. (Figure 2)

5. The reviewer asked for the information about indication for ECMO and types of ECMO (VA or VA modes). We revised the Table 2 and added the description in the text. The indication and diagnosis of patients needing ECMO support were shown in Table 2. Among 168 patients, 116 patients received veno-arterial ECMO and 52 patients received veno-venous ECMO. (Table 2, page 12, line 165-168)

Answers to Reviewer 2:

1.The review recommended for additional editing for English grammar throughout the manuscript. The manuscript had received English editing according to Editor’s suggestion.

2. The reviewer provided information about recent meta-analysis (PMID 31284451). We added this important reference and modified the introduction section. (page 4, line 50-52) 

3. The reviewer proposed concerns on whether the study is prospective or retrospective. Our study was retrospective cohort study, and we had modified our methods section and Figure 2 flowchart. (page 7, line 93-94)

4. The reviewer asked for information about types of ECMO. Among 168 patients, 116 patients received veno-arterial ECMO and 52 patients received veno-venous ECMO. We revised the Table 2 and added the description in the text. (Table 2, page 12, line 165-168)

5. The reviewer asked for the information about the duration of ECMO support and the differences between subgroups of patients. The median duration of ECMO support of all patients was 7 (IQR: 4-12) days, and AKD subgroup had longer duration of ECMO support (9 versus 6 days, p=0.024) than non-AKD subgroup. AKD stages were still significant independent predictors for survival after adjusting for duration of ECMO support. (Table 1, page 10, line 146; Table 3, page 13, line 181-183).

6. The reviewer proposed concerns on whether we recruited only patients with ECMO support having concomitant AKI. We are so sorry about the written mistake in methods section. Actually, we enrolled patients under ECMO support with AKI and without AKI. We modified manuscript and Figure 2 to describe a more clearly sequential story of renal failure in the form of a flowchart. (Figure 2, page 7, line 93-99)

7. The reviewer proposed concerns on “the highest serum creatinine value for staging AKI and AKD”. We adjusted Figure 2 and text to clarify the issue. We enrolled patients with ECMO support who survived more than 7 days with and without AKI. The index date was the first date of ECMO support. The highest serum creatinine value within 7 days was used for staging AKI and 7-90 days for staging AKD. (Figure 2, page 8, line 113-121)

8. The reviewer asked for the detail about how was baseline creatinine determined. Baseline serum creatinine was defined as lower value within 30 days before ECMO support, or defined as lower value within 30 days after ECMO support if no serum creatinine value was available within 30 days before ECMO support. (page 8, line 122-123; page 9 124-125)

9. The reviewer suggested that other variables would be useful to include in this report. We reviewed our data and the prevalence of baseline CKD was 28.6%. We added the information in the text. Other variable such as quantity of fluid, volume balance, and baseline medication were not available in our data source. We added these factors into our limitation section. (page 11, line 150-151; page 20, line 261-264)

10. The reviewer asked for the information of median survival time for AKD stage 0 (non-AKD) patients. More than half of the non-AKD patients survived 10 years after ECMO support, so the median survival time for non-AKD patients was more than 10 years. We added this information in the text. (page 14, line 186-187)

11. The reviewer proposed concerns on whether urine output, diuretics use, and volume balance influenced outcome. The urine output on ECMO first day was an independent predictor for survival in our study. AKD stages were also independent predictors for survival after adjusting urine output on ECMO first day or not. Volume status and diuretics use were not available in our data source. These were limitation of our study. We added these in the text. (page 20, line 263-266)

12. The reviewer proposed concerns on the impact of urine output or AKI on subsequent AKD. The reviewer also suggested using the eGFR (CKD-EPI equation) for confounder adjustment. Our data demonstrated urine output on first day of ECMO support was a significant independent risk factor for developing AKD after adjusting other variables. However, AKI was not a independent risk factor for subsequent AKD. We replaced serum creatinine value with eGFR (CKD-EPI equation) according to reviewer’s opinion. We modified Table 1, Table 3, Table 4, and the text. (page 8, line 115-116; page 11, line 166; page 12, line 167; page 15, line 193-195; page 16, line 205-208)

13. The reviewer asked for whether the presence of AKD is related to increased CKD, which thus translated to worse survival and would a patient who developed AKD that resolved without developing CKD possess the same worse outcomes. We modified Figure 2 to describe a sequential story of renal failure in the form of a flowchart. Seventy-seven patients died within 7-90 after ECMO support and 5 patients died 90 days after ECMO support. Among the patients who survived more than 90 days, only 19 patients developed CKD. Consequently, the majority of mortality could not attribute to complication of CKD. The patients who developed AKD that resolved without developing CKD seemed to have better outcome than patients with CKD. Among AKD patients who survived more than 90 days, 17 patients did not developed CKD and the mortality rate was 29.4%; 12 patients developed CKD and the mortality rate was 58.3%. However, the case number was too small to be further analyzed. (Figure 2)

14. The reviewer suggested stratified ECMO patients by primary diagnosis. However, this is a singer center study with a relative small case number. We had described this limitation in the text. (page 20, line 268-270)

15. The reviewer asked for the information of patients requiring RRT. Among 58 AKI stage 3 patients, 56 patients had received CRRT. (Figure 2)

16. The reviewer proposed concern on how pre-existing CKD cause progression of AKI to CKD. Patients with CKD have relatively small renal reserves, and they are vulnerable to develop renal progression after AKI insult which may be attributed to further increased single nephron glomerular pressure. (page 19, line 248)

17. The reviewer proposed concern on exclusion of patients who survived less than 7 days. Among initial 260 ECMO patients, 92 patients died within 7 days. We must exclude these patients because the diagnosis of AKD is based on 7-90 days after insult of renal injury. (Table 2)

18. The reviewer asked for the information of amelioration of AKD severity. We think it would be similar to management with patients with CKD but we can intervene in earlier period (7-90 days after insult of renal injury). Renin-angiotensin system blockers, sodium-glucose cotransporter 2 inhibitors, and anti-inflammatory agents may play a role.

19. The reviewer asked for the information about AKI biomarkers. Novel AKI markers such as NGAL were not checked in our study. So we may underestimate those patients having AKD stage 0. (page 20, line 261-263)

20. The reviewer asked for the different diagnosis effects of AKI, AKD and CKD with survival in ECMO patients. Our data demonstrated that AKI stage was independent predictors for 30-day survival and AKD was independent predictors for 10-year survival. However, only 19 patients who survived more than 90 days developed CKD. Consequently, our data is too small to analyze the effect of CKD on outcome. AKI and AKD may be used as short-term and long-term predictors respectively. (Figure 2)

Answers to Reviewer 3:

1.The reviewer proposed concerns on the inclusion criteria. We are so sorry about the written mistake in methods section. Actually, we enrolled patients under ECMO support with AKI and without AKI. We modified manuscript and Figure 2 to describe a more clearly sequential story of renal failure in the form of a flowchart. (Figure 2, page 7, line 93-99)

2. The reviewer asked for the time frame for assessment of AKD. The index date was the first day of ECMO support. AKD was defined 7-90 days after ECMO support. We modified Figure 2 to demonstrate the time frame for assessment AKD. (Figure 2)

3. The reviewer suggested deleted “10-year cohort study” in this manuscript. We adjusted our title and manuscript according to reviewer’s opinion.

4. The reviewer asked for the detail about how was baseline creatinine determined. Baseline serum creatinine was defined as lower value within 30 days before ECMO support, or defined as lower value within 30 days after ECMO support if no serum creatinine value was available within 30 days before ECMO support. (page 8, line 122-123; page 9 124-125)

5. The reviewer provided important recent study about the ECMO patients who develop AKI requiring RRT carry 3.7-fold higher hospital mortality (PMID:31284451). We added this reference and modified the introduction section. (page 4, line 50-52) 

6. The reviewer asked how did we obtain mortality outcome after hospital discharge. We linked patients’ ID to Taiwan’s National Health Insurance Research Database (NHIRD) to obtain the date of mortality.

7. The reviewer asked for the information of adjusting covariates selection. We selected basic demographic covariates including age, gender, baseline eGFR and other covariates that had p value > 0.2 for univariate analysis between two groups.

8. The reviewer proposed concerns on index date for survival analysis. We defined index date as the first day of ECMO support. (Figure 2)

9. The reviewer asked for language editing. We are very sorry for our poor English. The manuscript had received English editing according to reviewer’s opinion.

Answers to Reviewer 4:

1. The reviewer suggested describing a whole story of AKI to CKD. We modified Figure 2 according to reviewer’s valuable advice. The incidence of AKI was 78.8%. Among 168 patients, 58 patients developed AKI stage 3 and 56 patients received CRRT support with a median duration of 8 days. Among 122 patients with AKI, 62 patients developed AKD. Nineteen patients developed CKD 90 days later. Among 122 patients with AKI, 40 patients did not progress to CKD 90 days later. (Figure 2)

2. The reviewer asked for describing the incidence of AKI in the first seven days and its impact on outcomes including mortality. In our study, 205 patients developed AKI with incidence rate of 78.5%. The impact of AKI on mortality did not be mentioned in our manuscript because we excluded the patients who died within 7 days after ECMO support (N=92). We revised Figure 2 to describe a sequential story of renal failure and the number of exclusion in the form of a flowchart. (Figure 2)

3. The reviewer asked for evaluating the duration of renal injury. There were 205 patients with AKI (0-7 days), 75 patients with AKD (7-90 days), and 19 patients with CKD (>90 days). We revised Figure 2 to describe the detail of flowchart. (Figure 2)

4. The reviewer suggested using a flow diagram to demonstrate that how many patients with AKI developed AKD later. Among the the patients who survived more than 7 days after ECMO support (N=122), 62 patients developed AKD in the period of 7-90 days after ECMO support. We revised the Figure 2 according to reviewer’s valuable suggestion. 

5. The reviewer asked for the number of patients who developed CKD. A total of 19 patients developed CKD 90 days after ECMO support. (Figure 2)

6. The reviewer asked for the information of CRRT number and duration, and the practice for CRRT on ECMO. Among patients with AKI stage 3 who survived more than 7 days after ECMO support, 56 patients received CRRT. All patients received continuous veno-venous hemofiltration (CVVH). We separated vascular access and CRRT machine from ECMO circuit. Accordingly, additional vascular access points must be used. The method provided the advantage of no interference with ECMO hemodynamic while having the ultrafiltration controlled by the CRRT machine. 

7. The reviewer asked for the information of mortality in the whole cohort. Among 168 patients with ECMO support who survived more than 7 days, 82 patients expired in the 10-year follow-up. The mortality rate was high and we used many variables in the regression analysis for adjustment.

8. The reviewer suggested shortening the results section. We revised out results section according to reviewer’s opinion. (page 11, line 153-163)

9. The reviewer asked for the information of tests of normality. Kolmogorov-Simirnov method was used to test normality of numerical variables. Differences in continuous variables between the AKD group and non-AKD group were analyzed by Student t-test or Mann Whitney U test according to results of test of normality. (page 9 line 140-141; page 10, line 142)

Answers to Reviewer 5:

1. The reviewer suggested describing epidemiology on ECMO and AKI and providing the definitions of the AKD in the introduction section. A recent meta-analysis study reveals that among patients receiving ECMO, the incident rate of AKI is 62.8%. The AKD stage 1 was defined as an increase of serum creatinine to 1.5-1.9 times, stage 2 as an increase of serum creatinine to 2.0-2.9 times, and stage 3 as an increase of serum creatinine ≥ 3.0 times, from the baselines in the period of 7-90 days after an insult of renal injury. We revised the introduction section according to reviewer’s suggestion. (page 4, line 51-53; page 5, line 61-70)

2. The reviewer provided important recent study about the ECMO patients who develop AKI (PMID:31284451). We added this reference and revised the introduction section. (page 4, line 51-54) 

3. The reviewer suggested describing more data in the patient schema. We modified Figure 2 according to reviewer’s valuable advice. The incidence of AKI was 78.8%. Among 168 patients, 58 patients developed AKI stage 3 and 56 patients received CRRT support with a median duration of 8 days. Among 122 patients with AKI, 62 patients developed AKD. Nineteen patients developed CKD 90 days later. (Figure 2)

4. The reviewer asked for the detail about how was baseline creatinine determined. Baseline serum creatinine was defined as lower value within 30 days before ECMO support, or as lower value within 30 days after ECMO support if no serum creatinine value was available within 30 days before ECMO support. (page 8, line 122-123; page 9 124-125)

5. The reviewer asked for the information about admission serum Creatinine between AKD and non-AKD groups. The mean serum creatinine on first day of ECMO support were 2.2 and 2.5 mg/dL in non-AKD and AKD group, respectively. There was no statistically significant difference (P = 0.208).

6. The reviewer asked for the information about medications which could potentially influence the serum creatinine. Other variable such as quantity of fluid, volume balance, and baseline medications were not available in our data source. We added these factors into our limitation section. (page 11, line 150-151; page 20, line 261-264)

7. The reviewer asked for information about types of ECMO. Among 168 patients, 116 patients received veno-arterial ECMO and 52 patients received veno-venous ECMO. We revised the Table 2 and added the description in the text. (Table 2, page 12, line 165-168)

8. The reviewer proposed concerns on AKD staging for risk stratification in this critically ill patient. We focused on the impact of AKD on survival, so we must exclude the patient who expired within 7 days since ECMO support. We can consider intervening in this earlier period (7-90 days after insult of renal injury) via renin-angiotensin system blockers, fluid balance management, nephrotoxic agent avoidance. Further novel agents such as sodium-glucose cotransporter 2 inhibitors and anti-inflammatory agents may also play a role. 

Answers to Reviewer 6:

1. The reviewer provided important recent study about the pediatric patients with ECMO support who develop AKI . We added “Another previous published meta-analysis shows that incidence of AKI and severe AKI in the pediatric population requiring ECMO is high.” in the text. (page 4, line 54-56)

2. We deleted the sentence “The AKD is an emerging medical entity in the field of critical care and the clinical significance of this kidney condition remains unraveled to date.” according to reviewer’s suggestion.

3. We are so sorry about the spelling mistake of KDIGO. We have revised the manuscript according to reviewer’s opinion.

4. The reviewer suggested citing a reference about the burden of AKI. We added ”Global burden of AKI is around 13.3 million cases a year with hospitalizations for AKI rising over time. A published meta-analysis shows that in the United State alone there is one hospitalization associated with AKI every 7.5 minutes. ” in the introduction section according to reviewer’s suggestion.

5. The reviewer proposed concerns on the definition of AKI and AKD. We defined AKI according to KDIGO criteria in which renal failure developed within 7 days after ECMO support. AKD was defined as the clinical condition in which renal injury is present between 7-90 days after initiation of ECMO support. We modified Figure 2 to describe a sequential flowchart about study process. (Figure 2; page 8, line 116-124)

6. The reviewer proposed concerns on why AKI is not an independent predictor of AKD in our study. Among 92 patients who died within 7 days after ECMO support, 83 patients developed AKI. We must excluded the patient who died within 7 days because AKD was defined as renal injury within 7-90 after insult. A large portion of patients with AKI who died within 7 days after ECMO support may explain why AKI is not an independent predictor of AKD in our result.

7. Our manuscript had received English language editing according to reviewer’s suggestion.

---

## [Decision Letter · Decision Letter 1]

9 Mar 2020

PONE-D-19-31910R1

Acute Kidney Disease Stage Predicts Outcome of Patients on Extracorporeal Membrane Oxygenation Support

PLOS ONE

Dear Dr Chen,

Thank you for submitting your manuscript to PLOS ONE. After careful consideration, we feel that it has merit but does not fully meet PLOS ONE’s publication criteria as it currently stands. Therefore, we invite you to submit a revised version of the manuscript that addresses the points raised during the review process.

ACADEMIC EDITOR: Please adjust the manuscript according to Reviewer 2' comments and please provide figure 2 as was requested by multiple reviewers.

We would appreciate receiving your revised manuscript by Apr 23 2020 11:59PM. To enhance the reproducibility of your results, we recommend that if applicable you deposit your laboratory protocols in protocols.io, where a protocol can be assigned its own identifier (DOI) such that it can be cited independently in the future. For instructions see: http://journals.plos.org/plosone/s/submission-guidelines#loc-laboratory-protocols

We look forward to receiving your revised manuscript.

Kind regards,

Justyna Gołębiewska

Academic Editor

PLOS ONE

Reviewers' comments:

Reviewer's Responses to Questions

**Comments to the Author**

1. If the authors have adequately addressed your comments raised in a previous round of review and you feel that this manuscript is now acceptable for publication, you may indicate that here to bypass the “Comments to the Author” section, enter your conflict of interest statement in the “Confidential to Editor” section, and submit your "Accept" recommendation.

Reviewer #1: All comments have been addressed

Reviewer #2: (No Response)

Reviewer #3: (No Response)

Reviewer #5: All comments have been addressed

Reviewer #6: All comments have been addressed

2. Is the manuscript technically sound, and do the data support the conclusions?

Reviewer #1: Yes

Reviewer #2: Partly

Reviewer #3: (No Response)

Reviewer #5: Yes

Reviewer #6: Yes

3. Has the statistical analysis been performed appropriately and rigorously? 

Reviewer #1: Yes

Reviewer #2: I Don't Know

Reviewer #3: (No Response)

Reviewer #5: Yes

Reviewer #6: Yes

4. Have the authors made all data underlying the findings in their manuscript fully available?

Reviewer #1: Yes

Reviewer #2: Yes

Reviewer #3: (No Response)

Reviewer #5: Yes

Reviewer #6: Yes

5. Is the manuscript presented in an intelligible fashion and written in standard English?

Reviewer #1: Yes

Reviewer #2: Yes

Reviewer #3: (No Response)

Reviewer #5: Yes

Reviewer #6: Yes

6. Review Comments to the Author

Reviewer #1: I thank the author(s) for addressing my questions and concerns. Given the changes made and the major concerns being addressed, I have no further reservations regarding publication of the revised version of the manuscript.

Reviewer #2: I do think this article provides useful clinical information for patients who survive the 0-7 days and develop AKD, with the limitations noted.

I appreciate the English editing performed. I also appreciate the multiple changes made through the manuscript; the authors were responsive to the suggestions/criticisms.

Consider rewriting the definition of AKD. Currently, as it is written, “Stage 1 AKD is defined as an increase of serum creatinine level to 1.5-1.9 73 times, stage 2 AKD is defined as an increase of serum creatinine level to 2.0-2.9 times, and stage 3 74 AKD is defined as an increase of serum creatinine level to ≥ 3.0 times the baselines levels in 7-90 75 days after renal injury [11].” Individuals may read this as AKD patients are those who have an acute increase between 7-90 days, but patients who developed an AKI that persisted after the 0-7 days were excluded because it was not an acute increase during the 7-90 days. Instead, if the authors meant to write that AKD is defined as an increase of creatinine that occurred or persisted in the 7-90 days, it would better define the criteria for AKD and also the patient’s you are specifically studying.

Regardless, depending on the above, if there are patients who fit into AKD after they developed an acute creatinine rise in the 7-90 days (and not before in the 0-7 days), then one of the central assumptions of this paper is that the AKD was caused by etiologies primarily related to ECMO support, as AKD was defined as “a clinical condition characterized by renal injuries that occur between 7 and 90 days after initiation of ECMO support.” This would be a potential limitation as other insults may have occurred that caused the AKD in the 7-90 days unrelated to ECMO. For instance, if a patient developed AKI with Cr 2.0 and it improved to 1.2 by day 6 but then AKD with re-increase of Cr of 1.8 at day 10, this would potentially constitute a different patient with associated etiology of injury and associated risks of AKI/AKD from a patient who had no AKI with AKD Cr at day 10 or a patient who had AKI Cr 3.2 at day 2 and then creatinine 1.8 at day 10.

It bears mentioning that this is not a limitation not only of the author’s study, but many retrospective articles studying AKI/AKD. Patients without prior baseline creatinine before surgery had their baseline creatinine defined as the lowest following ECMO support. This will lead to misdiagnoses, and without knowing the volume of fluids, blood products, or other contributing factors, difficult to ascertain if the baseline creatinine is accurate or a dilution. Again, this is a challenge shared by many studies, and the authors noted this appropriately in the revised limitations section. I would ask the following two questions, and am unsure if the authors can answer: 1) How many patients had a baseline creatinine that was defined in the 30 days following ECMO support, and 2) why did you choose lowest creatinine to use as the baseline creatinine following ECMO rather than after surgery?

Since low urine output is a marker associated with kidney injury (used in the KIDGO and AKIN standard definition of AKI), it is not surprising to me that it is associated with mortality and AKD. Hence, I would have used it as a confounder for adjustment during analysis, rather focus on it as one of the main findings and significant of this study (I believe the authors did use urine output on day 1 for statistical analysis, but I’m not sure it is such an important finding that the discussion should start off by noting the association of UOP with the results). Instead, I would consider focusing more on the association of AKD with ECMO and overall mortality/survival, (especially since we only have urine output on one day, which is limited).

I do not see Figure 2 in the submitted revision. Hence, I was unable to assess for this revised change on the impact of my prior questions.

Page 14, line 187. May be useful to include more specific data (actual days rather than half a month for the manuscript portion). Could consider adding a table (if unable to due to journal’s limitations, consider supplemental).

I am unsure from a statistical perspective if it offers additional information, but as the reviwers commented that “More than half of the non-AKD patients survived 10 years after ECMO support, so the median survival time for non-AKD patients was more than 10 year,” then what was the average survival time?

It should be mentioned in the discussion or limitations that AKI was not found as an independent risk factor of subsequent AKD (even though it has been associated with CKD as noted by the authors), possibly due to drop out from mortality. My prior comment about “The reviewer proposed concern on exclusion of patients who survived less than 7 days. Among initial 260 ECMO patients, 92 patients died within 7 days. We must exclude these patients because the diagnosis of AKD is based on the renal function of 7-90 days after renal injury. (Table 2)” was perhaps a poor attempt at trying to think about how excluding these patients may have impacted your results One of the potential impacts as noted above.

Noted I do not think they answered one reviewer's question about duration of CRRT.

The authors replied to one suggestion, “The reviewer asked for the information of amelioration of AKD severity. We think it would be similar to management with patients with CKD but we can intervene in earlier period (7-90 days after renal injury). Renin-angiotensin system blockers, sodium-glucose cotransporter 2 inhibitors, and anti-inflammatory agents may play a role.” Without discussing in the manuscript the evidence or hypothesis of how interventions for AKD would improve outcomes, I would suggest softening the word “mandatory” for the statement “AKD staging should be mandatory for patients on ECMO support as it may help in risk stratification of critically ill patients” (page 20, line 278-279.

Reviewer #3: my comments have been addressed as much as possible. If author cannot. they acknowledged them in the limitations.

Reviewer #5: I have reviewed the manuscript before and made comments to which the authors have satisfactorily answered and modified the manuscript accordingly.

Reviewer #6: You have addressed all changes as specified. Unncessary sentences have been removed and new references and clarifications have been addressed appropriately.

7. PLOS authors have the option to publish the peer review history of their article (what does this mean?). If published, this will include your full peer review and any attached files.

Reviewer #1: No

Reviewer #2: No

Reviewer #3: No

Reviewer #5: No

Reviewer #6: Yes: Sohail Abdul Salim

---

## [Author Response · Author response to Decision Letter 1]

15 Mar 2020

Answers to Reviewer 2:

1. The reviewer suggested rewriting the definition of AKD. We revised the definition of AKD according to reviewer’s suggestion. (page 5, line 71, 75; page 8, line 121)

2. The reviewer suggested adding the limitation about other insults that cause renal injury occurring in the 7-90 days. We revised the section of discussion according to reviewer’s suggestion. (page 20, line 275-277)

3. The reviewer asked for the information about baseline serum creatinine. 42 patients had a baseline creatinine that was defined in the 30 days following ECMO support. (20 patients with ARDS, 5 patients with AMI, 7 patients with myocarditis, 1 patient with decompensated heart failure, 2 patients with shock status simultaneous with hypoxic respiratory failure, 3 patients with sudden collapse, 4 patients with severe traumatic injury). We choosed lowest creatinine to use as the baseline creatinine following ECMO rather than after surgery because only 79 patients who had received surgery in our study. (72 patients with post-cardiotomy cardiogenic shock, 4 patients with severe traumatic injury, 3 patients with post-lung or heart transplantation). 

4. The reviewer suggested focusing more on the association of AKD with mortality but not urine output. We revised the manuscript according to reviewer’s opinion. (page 18, line 231-232)

5. The reviewer did not see the revised Figure 2. We have re-uploaded the file of Figure 2.

6. The reviewer suggested replacing “half a month” with actual days. We revised the manuscript. (page 14, line 188).

7. The reviewer asked for the information about the mean survival time of non-AKD patients. The mean survival time of non-AKD patients was 88.5 months. 

8. The reviewer proposed concern that why AKI is not a risk factor of AKD in our study. We added “AKI is an independent risk factor of CKD. In our study, univariate analysis showed that AKI is a significant risk factor (HR: 2.623, p = 0.001) of AKD among patients receiving ECMO support. However, there is no statistical significance in multivariate analysis after adjusting for age, sex, duration of ECMO support, APACHE II score, SOFA score, GCS score, MAP, urine output, serum potassium level, and baseline eGFR (HR: 2.257, p = 0.133). The reason may be attributed to the relative small size of our study.” in the section of discussion. (page 19, line 247-252)

9. The reviewer asked for the information of duration of CRRT. The information was not available in this retrospective study. We added the limitation in the section of discussion. (page 20, line 273)

10. The reviewer suggested to add hypothesis of how interventions for AKD in the text and delete “mandatory” in the text. We added “The intervention of ameliorating AKD severity may be similar to management with patients with CKD. Renin-angiotensin system blockers, sodium-glucose cotransporter 2 inhibitor, and anti-inflammatory agents may play a role. Further studies are necessary to investigate the effects of increasing AKD severity on outcomes of patients on ECMO and the management of amelioration of AKD severity” in the section of discussion. We also softened the word “mandatory” according to reviewer’s suggestion. (page 19, line 259-263; page 21, line 287-288)

---

## [Decision Letter · Decision Letter 2]

25 Mar 2020

Acute Kidney Disease Stage Predicts Outcome of Patients on Extracorporeal Membrane Oxygenation Support

PONE-D-19-31910R2

Dear Dr. Chen,

We are pleased to inform you that your manuscript has been judged scientifically suitable for publication and will be formally accepted for publication once it complies with all outstanding technical requirements.

With kind regards,

Justyna Gołębiewska

Academic Editor

PLOS ONE

Additional Editor Comments (optional):

Reviewers' comments:

Reviewer's Responses to Questions

**Comments to the Author**

1. If the authors have adequately addressed your comments raised in a previous round of review and you feel that this manuscript is now acceptable for publication, you may indicate that here to bypass the “Comments to the Author” section, enter your conflict of interest statement in the “Confidential to Editor” section, and submit your "Accept" recommendation.

Reviewer #2: (No Response)

2. Is the manuscript technically sound, and do the data support the conclusions?

Reviewer #2: Yes

3. Has the statistical analysis been performed appropriately and rigorously? 

Reviewer #2: Yes

4. Have the authors made all data underlying the findings in their manuscript fully available?

Reviewer #2: Yes

5. Is the manuscript presented in an intelligible fashion and written in standard English?

Reviewer #2: Yes

6. Review Comments to the Author

Reviewer #2: The authors have adequately addressed my questions. Based on the additional figure provided and data, it would be interesting to have a future study that looked at the difference in outcomes between patients who developed AKI that persisted into AKD and those that did not have AKI but developed AKD, per the new defined definition of AKD. This might show differences in outcomes that may be clinically relevant, but I agree with the authors that this was neither the intent of this study and the number of patients is insufficient to study that on this study (e.g. no changes needed).

I woudl like to make one last comment that the reason I asked the authors further expound on how interventions for AKD would be taken is that use of sodium glucose 2 transport inhibitors is currently only recommended in a select patient population, not all CKD patients. Furthermore, it is unclear what anti-inflammatory agents they are referring to, as NSAIDs are contraindicated for CKD patients and would not be expected. Corticosteroids are also not recommended for routine CKD. Hence, I was attempting to prompt further details as readers may question this statement as it currently is.

Otherwise, as previously noted above, I would support publication.

7. PLOS authors have the option to publish the peer review history of their article (what does this mean?). If published, this will include your full peer review and any attached files.

Reviewer #2: No

---

## [Editor Report · Acceptance letter]

26 Mar 2020

PONE-D-19-31910R2 

Acute Kidney Disease Stage Predicts Outcome of Patients on Extracorporeal Membrane Oxygenation Support 

Dear Dr. Chen:

I am pleased to inform you that your manuscript has been deemed suitable for publication in PLOS ONE. Congratulations! Your manuscript is now with our production department. 

With kind regards,

on behalf of

Dr. Justyna Gołębiewska 

Academic Editor

PLOS ONE